# Oligonucleotide Therapies in the Treatment of Arthritis: A Narrative Review

**DOI:** 10.3390/biomedicines9080902

**Published:** 2021-07-27

**Authors:** Susanne N. Wijesinghe, Mark A. Lindsay, Simon W. Jones

**Affiliations:** 1MRC-ARUK Centre for Musculoskeletal Ageing Research, Institute of Inflammation and Ageing, University of Birmingham, Birmingham B15 2TT, UK; s.n.wijesinghe@bham.ac.uk; 2Department of Pharmacy and Pharmacology, University of Bath, Claverton Down, Bath BA2 7AY, UK; M.A.Lindsay@bath.ac.uk

**Keywords:** osteoarthritis, rheumatoid arthritis, synovitis, cartilage, bone, antisense, oligonucleotides, therapeutics

## Abstract

Osteoarthritis (OA) and rheumatoid arthritis (RA) are two of the most common chronic inflammatory joint diseases, for which there remains a great clinical need to develop safer and more efficacious pharmacological treatments. The pathology of both OA and RA involves multiple tissues within the joint, including the synovial joint lining and the bone, as well as the articular cartilage in OA. In this review, we discuss the potential for the development of oligonucleotide therapies for these disorders by examining the evidence that oligonucleotides can modulate the key cellular pathways that drive the pathology of the inflammatory diseased joint pathology, as well as evidence in preclinical in vivo models that oligonucleotides can modify disease progression.

## 1. Introduction

### 1.1. Rheumatoid Arthritis

RA is one of the most common chronic inflammatory conditions, with an estimated prevalence of up to 2% in some ethnic groups [1]. Largely a result of inflammation to the synovial lining (synovitis) of the joint, RA causes joint pain, stiffness, swelling and increased fatigue and results in increased disability and reduced quality of life. Furthermore, around a third of patients are unable to work within 2 years of diagnosis, thus causing a substantial socioeconomic burden [2,3,4]. There are currently several approved pharmacological drugs for the therapeutic management of RA patients [5]. Corticosteroids and non-steroidal anti-inflammatory drugs such as diclofenac can be administered locally to the painful joint by topical application to reduce joint pain and swelling [6]. However, these drugs do not modify the disease process and are also associated with adverse side effects over the long term when taken chronically. For example, glucocorticoids, when taken chronically, can lead to increased bone resorption [7] and skeletal muscle atrophy [8]. Importantly, the aim of current RA therapies is therefore to target the underlying disease pathology, a “treat-to-target” strategy to influence the disease course, not simply to reduce symptoms [9]. To this end, a number of disease-modifying anti-rheumatic drugs (DMARDs) have been approved for clinical use, such as methotrexate, hydroxychloroquine, sulfasalazine and leflunomide, which are immunomodulatory drugs [10]. For example, methotrexate inhibits leukotriene B4 synthesis by neutrophils, and suppresses the production of the pro-inflammatory cytokines IL-1, IL-6 and IL-8 [11,12]. Leflunomide inhibits pyrimidine synthesis, thus reducing lymphocyte proliferation [13], whilst hydroxychloroquine reduces immune responses [14] by inhibiting the toll-like receptor signalling [15]. The precise mechanism of action of sulfasalazine is not fully understood [16], but it is known to inhibit NF-kB activity and thus the production of inflammatory cytokines including tumour necrosis factor alpha (TNF-α) [17]. In recent years, improved understanding of the molecular pathology underlying RA disease has led to a revolution in RA treatment [18], with the emergence of a number of biological drugs designed to target specific disease-associated inflammatory cytokines (most notably TNF-α), and drugs targeting the activity of specific immune cell populations. Monoclonal antibodies targeting TNF-α, including adalimumab (Humira), etanercept and infliximab, have all demonstrated efficacy in reducing disease activity scores (DAS) in patients and in maintaining the disease in remission [19,20,21], as determined by the number of swollen and painful joints and biomarkers of inflammation; thus, reducing the risk of irreversible damage to joints. Similar disease modification has been achieved by biologics targeting other inflammatory immune processes. For example, tocilizumab, a monoclonal antibody targeting the pro-inflammatory IL6 [22], rituximab, a monoclonal antibody targeting CD20 on B cells [23] and abatacept, a fusion protein composed of the Fc region of the immunoglobulin IgG1 fused to the extracellular domain of CTLA-4, which inhibits antigen-presenting cells from providing the co-stimulatory signal required for T-cell activation [24]. However, despite these successes, all these immunomodulatory medications are associated with adverse side effects and an increase in the risk of developing serious infections, such as pneumonia, due to their action in modulating the immune system [25,26]. Furthermore, with chronic treatment, efficacy can be greatly diminished due to the patient developing anti-drug neutralising antibodies [27]. Therefore, there is still a great unmet medical need to develop more effective and safer RA therapeutics.

### 1.2. Osteoarthritis

OA is the most common degenerative joint disorder in the world, affecting 303 million people globally [28]. Historically, OA was seen as a disease solely of the articular cartilage. However, it is now widely accepted that OA is a disease of the whole joint, including not only the loss of articular cartilage mass and meniscal and/or ligament damage, but also subchondral bone sclerosis and inflammation of the synovium and infrapatellar fat pad [29,30,31,32].

Similarly to RA, OA results in joint pain and increased stiffness, which leads to progressive disability and a reduced quality of life, amounting to a huge socioeconomic burden. Unfortunately, since age is a significant risk factor for OA, the prevalence of OA is set to increase in developed countries with our increasingly ageing society [33]

Unfortunately, current treatment options for patients are limited. Despite a number of clinical trials, including with biologics targeting pro-inflammatory cytokines such as adalimumab and tocilizumab [34], there are no pharmacological drugs that have been proven to modify the disease course of OA (DMOADs) [35]. Additionally, although platelet-rich plasma and glucosamine-based supplementation show some promise for alleviating pain and delaying OA progression, these effects have been highly variable and debated in trials [36,37]. Therefore, patients are advised to take non-steroidal anti-inflammatory drugs such as ibuprofen, which have limited efficacy and are associated with toxicity over the long term. Patients with knee OA can receive intra-articular corticosteroid injections into the diseased joint to alleviate pain and inflammation [27]. However, the benefit for patients is highly unpredictable, with ~30% of knee OA patients reporting little to no improvement in symptomatic pain [38]. Furthermore, repeated steroid injections into the joint can exacerbate cartilage damage. As a result, OA patients live with painful symptoms for several years before undergoing a joint replacement surgery, a procedure with a high percentage of patient dissatisfaction. A particular challenge in developing a DMOAD has been the effective delivery of drugs into the cartilage, which as an avascular tissue, is not highly amenable for conventional drug delivery. In order to target the chondrocyte cells that mediate cartilage OA pathology, such drugs need to be able to penetrate the full depth of the cartilage tissue [39]. Furthermore, the focus for many years on identifying candidate targets that directly mediate cartilage degradation, such as the matrix metalloproteases (MMPs), neglected the important role of the other tissues in the joint in driving OA disease progression, such as the synovium and subchondral bone. To this end, the development of oligonucleotides, that modulate expression of targets at an upstream level, where functional protein products are not made, may offer the potential to target pathological processes across multiple cell types, including chondrocytes in the cartilage, osteoblasts in the bone and fibroblasts in the synovium.

In this review, we discuss the potential for the development of oligonucleotide therapies in both RA and OA joint disease by examining the evidence for oligonucleotide therapies to modulate disease pathology and disease-associated cellular pathways within the multiple tissues of the joint (Figure 1).

## 2. Methods

Studies used to write this narrative review were collected from the PubMed database. Search criteria for original articles included key words “oligonucleotide”/“siRNA”/“LNA” and “osteoarthritis” and/or “rheumatoid arthritis”, and we filtered for studies between 2000 and the present. Studies were deemed relevant if oligonucleotides were demonstrated to alleviate inflammation, pain and ameliorate OA and/or RA disease progression. The details of those studies included are summarised in Table 1 and Table 2.

## 3. Oligonucleotides Targeting Synovial Inflammation

Inflammation of the synovial membrane (synovitis) is a key hallmark of both RA and OA joint disease. Fibroblast-like synoviocytes (FLS), also termed synovial fibroblasts, become activated and hyperplastic in RA [74,75]. Through the infiltration and subsequent interaction of immune cells in the synovium (including leukocytes and resident macrophages), synovial fibroblasts contribute to a sustained chronic inflammatory state within the joint by releasing pro-inflammatory cytokines into the synovial joint fluid, such as TNFα and IL-6 [75,76], which also promotes bone resorption. Synovitis and the associated bone erosion are detected by magnetic resonance imagining (MRI) of the RA joint [77]. Similar to RA, synovitis is now widely recognised to play a significant role in OA joint pathology, with synovitis evidence by ultrasound, MRI and histopathology prior to radiographic signs of cartilage damage, with increased infiltration of activated B and T cells and synovial proliferation and hypertrophy [34,78,79,80,81]. Synovial fibroblasts from OA patients are more inflammatory compared to non-diseased patients and secrete greater levels of pro-inflammatory cytokines IL-6 and IL-8 [65].

In attempting to develop oligonucleotide therapies to target synovitis, modulation of the activated proliferative inflammatory synovial fibroblast phenotype using antisense oligonucleotides has been documented. For example, Nakazawa et al. reported that antisense oligonucleotides targeting Notch-1 protein inhibited both basal and TNFα-induced proliferation of human synovial fibroblasts isolated from either RA or OA patient synovium [66], whilst antisense knockdown of the gene PTPN11, which encodes SHP-2 (a known proto-oncogene), was reported to inhibit migration and survival of RA synovial fibroblasts [67]. More recently, susceptibility of human OA and RA synovial fibroblasts to fas-mediated apoptosis was increased by antisense oligonucleotides targeting the anti-apoptotic gene FLICE-inhibitory protein (FLIP) [60], and increased apoptosis of human RA synovial fibroblasts was induced upon antisense oligonucleotide targeting of galectin-9 [62]. These targets, unsilenced, provide protection against apoptosis, thus maintaining fibroblast populations contributing to persistent inflammation, and as such may be valuable targets in combatting synovial inflammation and hyperplasia. Furthermore, oligonucleotides have been demonstrated to reduce the inflammatory phenotype of activated OA and RA synovial fibroblasts. For example, the inflammatory fibroblast phenotype mediated by leptin, an adipokine found elevated in the synovial fluid of both OA and RA patients [35,82], was inhibited by antisense oligonucleotides targeting the leptin receptor (ObR), which reduced leptin-mediated IL-8 secretion [63] and IL-6 expression in OA fibroblasts [64]. Increasing evidence has emerged that long non-coding RNAs (lncRNAs), such as MALAT1, are central regulators of the inflammatory response [83]. LncRNAs are a relatively novel class of non-coding RNAs, which have been shown to regulate gene expression at both the epigenetic pre-transcriptional and post-transcriptional level through their ability to act as scaffolds for the binding of proteins and other RNAs [84,85,86,87,88]. Recently, the MALAT1 lncRNA was found to regulate the inflammatory response of articular OA chondrocytes [89] and synovial fibroblasts [65]. A locked nucleic acid (LNA) oligonucleotide targeting the MALAT1 lncRNA was found to inhibit both the proliferative and inflammatory phenotype of obese OA synovial fibroblasts [65]. Therefore, oligonucleotide targeting of lncRNAs that are dysregulated in the tissues of the arthritic joint could provide novel therapeutic strategies to target the epigenetic drivers of joint inflammation [87].

As yet, few studies have reported the effect of oligonucleotides designed to target the synovium in preclinical in vivo models of either RA or OA disease [59,61,90,91,92,93]. However, in a surgically induced experimental model of OA, a 21-mer end-capped phosphorothioate antisense directed against Dickkopf-1 (DKK1), the canonical Wnt pathway inhibitor, was intraperitoneally administered at a dose of 20 μg/kg/week for up to 12 weeks in rats and was found to ameliorate synovial vascularity [59]. Similarly, intra-articular injection of FoxC1 siRNA, which is a promoter of TNF-α and IL-8 production in synovial fibroblasts, was found to reduce arthritis development in rodent models of OA and RA [61,90].

## 4. Oligonucleotides Targeting Subchondral Bone Pathology

In bone homeostasis, receptor activator of nuclear factor kappa B (RANK)/RANKL pathway activates NF-kB-induced transcription factors that provide the balance between bone resorption and bone formation [94,95]. However, this homeostasis is lost in the pro-inflammatory microenvironment within the RA joint, where there is the promotion of bone erosion via the pro-inflammatory cytokine induction of RANKL, which binds to the RANK receptor on osteoclasts and activates their bone resorbing activity [94]. Conversely, in OA the subchondral bone becomes sclerotic, with pronounced trabecular thickening, areas of subchondral bone that are under-mineralised and the formation of bony spurs (osteophytes), which are notable in X-ray radiographs of the joint [35,94,96,97,98]. An often overlooked pathological feature in OA, there is now evidence that subchondral bone changes in OA precede and drive cartilage damage [99,100], with OA bone and OA osteoblasts exhibiting an abnormal type I collagen alpha1 homotrimer phenotype [29] with impaired mineralisation [101]. In human primary osteoblasts, ObR antisense oligonucleotides abolished the leptin-mediated production of oncostatin M [56], which, as a member of the IL6 family, has been associated with bone remodelling and cartilage volume loss in OA and RA [102] as well as ObR itself being associated with biomarkers of cartilage loss and bone remodelling over 2 years in knee OA patients [103]. In vitro, antisense oligonucleotides have also been shown to effectively impact on osteogenic differentiation [57]. In mesenchymal stem cells isolated from patients with knee OA, oligonucleotide antisense against miR29a impaired Wnt-mediated osteogenic differentiation via reducing Wnt3 expression [57].

In vivo, in either a surgically induced OA model anterior cruciate ligament (ACL) transection or a collagenase-induced OA model, end-capped DKK1 antisense oligonucleotides delivered intraperitoneally (10–50 µg/kg/week for up to 8 weeks) lowered disease severity, with a reduction in bone mineral density loss, reduced serum levels of bone resorption markers osteocalcin and CTX-1 and suppressed expression of TNF-α, IL-1β, MMP3 and RANKL [55]. Such findings are consistent with the known role of DKK1 in bone homeostasis as an endogenous inhibitor of the Wnt/beta-catenin signalling pathway. DKK1 is implicated in bone development, the pathological remodelling of bone in both OA and osteoporosis and mediating inflammation-induced bone loss by inhibiting osteoblast differentiation [104,105]. In osteoporosis patients, serum levels of DKK1 are negatively associated with bone mineral density in the femoral head and lumbar spine [106]. More recently, intra-articular injection of an siRNA oligonucleotide targeting YAP, which promotes osteogenesis and bone remodelling, reduced the aberrant subchondral bone formation in the ACL mouse model of OA [54]. Amelioration of sclerotic subchondral bone formation, as well as an overall reduction in OA severity score, has also been achieved in the mouse DMM model by intra-articular delivery of a 2′OME 5′Chol-modified antisense oligonucleotide (2 nmol) targeting the thyroid hormone receptor (THR) [58].

## 5. Oligonucleotides Targeting Cartilage Degeneration

Cartilage degradation in OA is a key hallmark of OA incidence and of disease progression. This is driven largely by a pathological switch in the phenotype of chondrocytes [35]. In healthy adult cartilage, the chondrocytes are embedded in the extracellular matrix and are in a relatively metabolically inactive state, where they produce type II collagen and aggrecan proteoglycans that hydrate the cartilage and provide the cartilage with its load-absorbing properties [107]. However, in OA the chondrocytes proliferate and become hypertrophic, switching from producing extracellular matrix proteins to producing catabolic MMPs and aggrecanases ADAMTS4 and ADAMTS5, which degrade type II collagen and aggrecan proteoglycan, respectively [35,107]. Induction of MMPs and aggrecanases is promoted by the inflammatory microenvironment of the joint [52], with pro-inflammatory cytokines secreted by various joint tissues and damage-associated molecular patterns (DAMPs) such as fibronectin fragments from degraded cartilage potentiating a vicious cycle of inflammation and cartilage damage via activation of TLRs, the MAPK pathway and the NF-kB pathway [35]. The cellular cross-talk between synovial fibroblasts and chondrocytes is fundamental in this, with IL-6 released from cartilage chondrocytes capable of binding to the soluble IL-6 receptor (sIL-6R) in the synovial fluid and this IL-6/sIL-6R complex transactivating the membrane-bound gp130 on fibroblasts to promote further IL-6 secretion [30]. This chondrocyte–fibroblast crosstalk is further exacerbated in obese patients with OA, where the adipokine leptin stimulates greater IL-6 secretion from articular chondrocytes [30].

In vitro, antisense oligonucleotides have been shown to modulate the inflammatory and catabolic phenotype of human OA chondrocytes. Targeted knockdown of the G-Protein Coupled Receptor (GPCR) RDC1 in human knee OA chondrocytes using antisense oligonucleotides modified the OA chondrocyte phenotype, with reduced expression of a panel of MMPs and hypertrophic markers [53]. Similarly, antisense oligonucleotides targeting either p38 MAPK or the downstream MAPKAPK2 (MK2), which regulates TNF stability via TPP regulation, inhibited IL-1β-induced production of MMP3, MMP13 and PGE2 [34,52]. Modulation of MMP13 expression in human chondrocytes was also reported using c-Fos and c-Jun antisense oligonucleotide, which acted to inhibit the potentiating action of SDF-1alpha on MMP-13 promoter activity [41]. In vivo, intra-articular injection of siRNA antisense targeting MMP13 and ADAMTS5, either alone or in combination, improved histological scores of OA severity in a murine DMM model, compared to non-targeting control siRNA [40].

In recent years, the profiling of non-coding miRNAs has identified a number of miRNAs that are dysregulated in OA diseased cartilage [45], and/or associated with disease progression [108,109], and several studies have now demonstrated modulation of the OA chondrocyte phenotype by antisense oligonucleotide targeting of these miRNAs. In vitro, antisense oligonucleotides against miR-320a reduced the IL-1β-mediated release of MMP13 in human chondrocytes [49], whilst conversely, miRNA oligonucleotide mimics of miR-98 and mir-146 reduced TNF-α and MMP13 production in human OA chondrocytes [45]. In vivo, miR-128-targeting antisense oligonucleotides administered by intra-articular injection were reported to slow articular cartilage degradation, reduce synovitis and slow subchondral bone changes in the ACL experimental model of OA at 8 weeks [43]. In the destabilisation of the medical menisci (DMM) experimental model of OA, miR-181a-5p LNA antisense oligonucleotides delivered by intra-articular injection (3 µL of 1 µg/µL per knee joint) attenuated cartilage destruction [46]. Similarly, intra-articular injection of antagomir-21-5p significantly attenuated the severity of OA in the DMM model, via modulating expression of FGF18 [47]. Furthermore, intra-articular injection of antisense oligonucleotides targeting miR-34a-5p was chondroprotective in a murine DMM model and in a high-fat diet/DMM model [50], whilst LNA antisense inhibition of miR-449a via intra-articular injection (100 nM injections twice weekly for 8 weeks) promoted cartilage regeneration and expression of type II collagen and aggrecan in a rat acute cartilage defect model after 4 and 8 weeks post surgery and in a rat DMM model [51].

In addition to miRNA inhibition, miRNA oligonucleotide mimics have also been demonstrated to have potential as arthritis therapeutics in vivo. A miRNA mimic of miR-26a/26b was found to promote chondrocyte proliferation in vitro by targeting of fucosyltransferase 4 (FUT4) and to attenuate development of OA in a rat ACL model of OA when delivered by single intra-articular injection at a dose of 5 nmol, 1 week after surgical induction of OA [48]. Similarly, intra-articular injection of a miR-145 mimic (at a dose of 50 µM twice a week for 7 weeks) reduced cartilage degradation in a rat DMM model via suppression of MKK4-mediated induction of TNF-α [44]. There is therefore increasing evidence that oligonucleotide therapeutics, when delivered by intra-articular injection, are capable of modulating the diseased inflammatory phenotype of arthritis cartilage.

## 6. Optimisation of Oligonucleotide Method of Delivery into Joint Tissues

One of the central challenges in the development of drugs to treat inflammatory joint diseases such as OA and RA is the effective delivery of the therapeutic to the diseased tissues of the joint. In particular, the articular cartilage represents a formidable challenge since it is avascular, and thus unmodified drug delivery into the chondrocytes occurs slowly and inefficiently [110]. Oligonucleotide therapeutics, at approximately 20 bases, are larger than most small-molecule drugs and are also anionic, thus making diffusion across a negatively-charged cell membrane challenging [110,111].

However, the ability to deliver therapeutics, including oligonucleotide therapeutics, directly into the joint space via intra-articular injection, thus maximising the ability of the oligonucleotides to target the diseased tissues, is an advantage. Indeed, the increasing number of studies demonstrating the efficacy of antisense oligonucleotides and miRNA mimics in reducing disease severity in experimental models of arthritis following intra-articular injection delivery is highly promising. Furthermore, delivery of the therapeutic locally to the joint limits the risk of unwanted side effects at distal sites [110,111]. Intra-articular delivery is particularly appealing for OA disease. In contrast to RA, OA usually affects only one or two large joints, commonly the knee or hip joints, with systemic inflammation largely absent.

### 6.1. Viral Delivery Systems in Inflammatory Joint Disease

In attempting to optimise both the stability and the delivery of oligonucleotide therapeutics in the joint, a number of strategies have been employed. For example, intra-articular injection provides a route for the local delivery of viral vectors (such as lentiviruses, adenoviruses, adeno-associated viruses) capable of carrying oligonucleotide cargo without the risk of systemic side effects [112]. In a murine experimental model of arthritis, lentiviral delivery of an antisense oligonucleotide targeting ATG12, an autophagy regulator, into the cartilage tissue by intra-articular administration was reported by Lian et al. [43]. Similarly, adenovirus delivery of miR-101 antisense oligonucleotides was shown to prevent cartilage degeneration in a chemically induced rat model of OA [42]. However, clinically there is still concern about whether viral-mediated delivery of therapeutics will lead to toxicity. Furthermore, viral-mediated delivery is costly. Therefore, non-viral delivery systems may represent a more rewarding therapeutic approach for the delivery of oligonucleotides.

### 6.2. Biomaterial Delivery Systems in Inflammatory Joint Disease

One alternative to viral delivery systems is the use of biomaterials, such as hydrogels. Hydrogels have been successfully used to sustain oligonucleotides within the microenvironment of the joint and to facilitate slow release. For example, a hyaluronic acid hydrogel was previously utilised to facilitate the controlled release of an LNA Gapmer oligonucleotide targeting PTGS2 (COX2), the gene which promotes the expression of the inflammatory pain mediator PGE2. Notably, the hydrogel-encapsulated COX2 LNA was shown to produce gene silencing in OA chondrocytes over a 14 day period [68]. Using a similar hydrogel approach, efficient knockdown of the aggrecanase ADAMTS5 was achieved up to 14 days in OA chondrocytes [69]. These data provide evidence that hyaluronic acid hydrogel delivery systems have the potential to provide the necessary bioavailability within the joint cavity to support uptake of oligonucleotide therapeutics into the joint tissues, including the avascular cartilage tissue.

### 6.3. Bioconjugation and Cell-Penetrating Peptide Delivery

Cell-penetrating peptides (CPPs) such as TAT, penetratin and antennapedia are short cationic peptide sequences that can translocate across cell membranes [113,114]. Therefore, a number of studies have demonstrated the utility of CPPs to transport biological cargo into cells in vitro and into tissues in vivo, including oligonucleotide cargo. For example, siRNAs targeting p38 MAPK conjugated to TAT or penetratin CPPs could deliver siRNA targeted knockdown in a mouse fibroblast cell line. However, in vivo these CPP conjugates provoked the innate immune response. Instead, bioconjugation of the siRNA to cholesterol was found to deliver p38 MPAK knockdown in lung tissues, following intra-tracheal delivery [70]. Bioconjugated cholesterol siRNAs have recently been shown to be efficacious in knocking out the expression of myostatin (a negative regulator of muscle mass) in mice following systemic delivery [71]. An alternative conjugation is the GalNAc (N-acetylgalactosamine) modification [115]. In particular, this modification has been shown to facilitate uptake into the liver following systemic administration, in part likely due to the high expression of the ASialoGlycoProtein Receptor (ASGPR) in hepatocytes, to which GalNAc has binding affinity for. As a result, GalNAc-conjugated antisense oligonucleotides targeting liver disease are currently in development [116]. However, since the GalNAc modification has been shown to confer metabolic stability to oligonucleotide conjugates, there are now additional GalNAc antisense oligonuclelotide conjugates in development for other conditions, including cardiovascular disease, hepatitis C and type 2 diabetes [117]. As yet, there are no published reports on the anti-inflammatory efficacy of GalNAc-modified oligonucleotides in vivo in the joint.

### 6.4. Nanoparticle Delivery Systems in Inflammatory Joint Disease

An alternative approach is the use of nanoparticles. Although confirmation in the clinical setting is still required, such nanoparticulated drugs (nanodrugs) have recently gained recognition in drug delivery for a number of therapeutic areas [118,119,120]. Indeed, Geiger et al. showed that nanoparticles carrying IGF-1 successfully penetrated bovine cartilage within 2 days of application [121]. Notably, nanoparticles can exploit the synovial capillary networks within the synovial tissue and previous studies have demonstrated that even systemic delivery of nanodrugs can be detected in the synovial joint in experimental models of RA [122]. In OA, since synovial tissue has fewer blood capillaries, intra-articular delivery of nanoparticle-decorated oligonucleotides is likely to still be required in order to achieve sufficient bioavailable delivery to the synovial joint tissues. Importantly, it has been shown that when cationic nanoparticles are administered by intra-articular injection, they can efficiently penetrate cartilage tissue [123,124,125,126]. To this end, intra-articular injection of polyethylene glycol (PEG) chain-modified single-walled carbon nanotubes (SWCNTs) successfully delivered morpholino antisense oligonucleotides into chondrocytes of both healthy and OA cartilage in mice, with no major adverse effects [72]. Furthermore, it was recently reported that a miR-141/200c aptamer-decorated nanoparticle persisted for 5 weeks in chondrocytes and provided chondroprotection when delivered by intra-articular injection into a DMM murine experimental OA model via inhibition of the IL-6/STAT3 pathway [73].

## 7. Conclusions

Oligonucleotide therapeutics represent an emerging but highly promising class of therapeutics to treat inflammatory joint disease. Although yet to be successfully tested in clinical trials for arthritis treatment, data from preclinical experimental models of arthritis provide evidence that the intra-articular delivery of oligonucleotides can modify OA disease pathology, by reducing synovitis, preventing sclerotic bone formation and protecting from cartilage damage. Importantly, since oligonucleotide therapeutics are based on gene sequences, they are expected to act specifically on the target gene, and thus may be considered less likely to have off-target effects and to elicit adverse side effects. Indeed, the same basic backbone chemistry of oligonucleotides and the safety profile of oligonucleotide therapeutics in the clinic provide confidence that late-stage clinical trial failure with this class of drug may be less common than with small-molecule or monoclonal antibody entities. Furthermore, they can be rapidly and cheaply synthesised, thus greatly reducing drug development timelines. As with conventional drug delivery, approaches to optimise the stability and delivery efficacy of oligonucleotide therapies into the joint tissues remain a major area of current focus and barrier to effective disease modification. To this end, the use of nanocarriers and the GalNAc bioconjugation represent promising approaches, but further clinical studies will provide important information on the safety profile of such carriers in the clinical setting for patients with inflammatory joint disease.

## Figures and Tables

**Figure 1 biomedicines-09-00902-f001:**
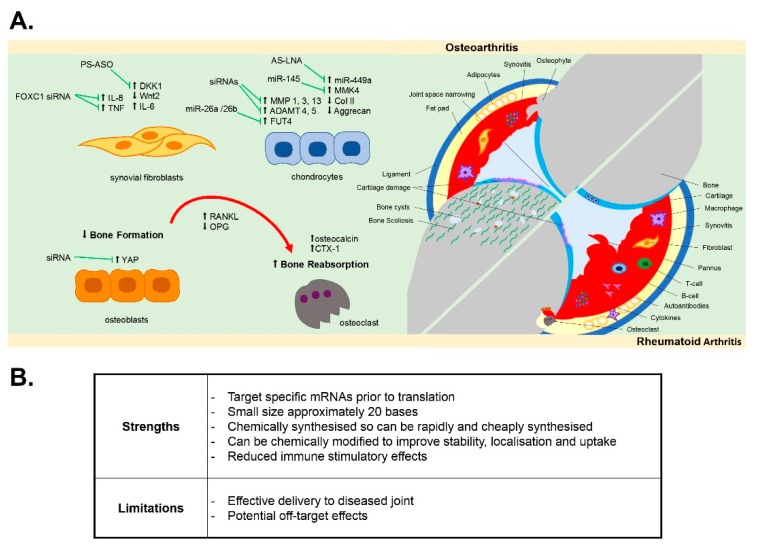
Features of OA and RA pathology and in vivo oligonucleotide targets. (**A**) RA is an autoimmune condition driven by inflammation of the synovial lining which degrades cartilage and activates osteoclast bone reabsorption, whilst OA is a degenerative joint disorder involving the loss of articular cartilage mass, ligament damage, subchondral bone sclerosis and fat pad and synovial inflammation. Inflammatory synovial fibroblasts, along with other joint cells, produce pro-inflammatory cytokines which stimulate RANKL production in osteoblasts, thus promoting osteoclastogenesis and bone resorption. Damaged cartilage exacerbates synovial inflammation, driving further cartilage loss. (**B**) Summary of the strengths and limitations of using oligonucleotides as therapeutics. ADAMTS 4, 5: a disintegrin and metalloproteinase with thrombospondin motifs 4, 5, AS-LNA: antisense locked nucleic acid, BMP: bone morphogenic proteins, Col II: collagen type 2, CTX-1: cytotoxin-1, DKK1: Dickkopf-elated protein 1, FOXC1: forkhead box C1, FUT4: fucosyltransferase 4, IL-6, 8: interleukin-6, 8, miR-26a, 26b, 145, 449a: microRNA-26a, 26b, 145, 449a, MMK4: mitogen activated protein kinase, MMP-1, 3, 13: matrix metalloproteinase-1, 3, 13, OA: osteoarthritis, OPG: osteoprotegerin, PS-ASO: phosphorothioate antisense oligonucleotide, RA: rheumatoid arthritis, RANKL: receptor activator of nuclear factor kappa-Β ligand, siRNA: small interfering RNA, TNF: tumor necrosis factor, Wnt2: Wnt family member 2, YAP: yes-associated protein.

**Table 1 biomedicines-09-00902-t001:** Efficacy of oligonucleotides in OA and RA preclinical models.

Cartilage					
**Target**	**Type of Oligonucleotide**	**Target Cell/Tissue**	**Study Model**	**Function**	**Ref.**
ADAMTS5	siRNA	cartilage	in vivo—MM mouse	Silencing alone or in combination with MMP13 siRNA improved histological scores of OA severity.	[40]
c-Fos	antisense oligonucleotide	chondrocytes	in vitro	Knockdown acted to inhibit the potentiating action of SDF-1α on MMP-13 promoter activity.	[41]
c-Jun	antisense oligonucleotide	chondrocytes	in vitro	Knockdown acted to inhibit the potentiating action of SDF-1α on MMP-13 promoter activity.	[41]
miR-101	adenovirus-miRNA	cartilage	in vivo—MIA rat	Silencing by intra-articular injection reduced cartilage degeneration in an OA model. Microarray analysis found downregulation of several cartilage-related genes.	[42]
miR-128	antisense oligonucleotides	chondrocyte	in vivo—ACLT rat	Intra-articular injections silencing miR-128a reduced cartilage degradation, synovitis and subchondral bone damage in the ACLT rat model via Atg12.	[43]
miR-145	miRNA mimics	cartilage	in vivo—DMM rat	Intra-articular injection of miRNA mimics reduced cartilage degradation via suppression of MKK4, which negatively regulates TNFα-mediated JNK, p38, p-c-Jun and p-ATF2, thus repressing MMP3, MMP13 and Adamts-5.	[44]
mir-146	pre-miRNA mimics	chondrocytes	in vitro	Overexpression significantly attenuated IL-1β-induced reduced TNFα production.	[45]
miR-181a-5p	locked nucleic acid	chondrocyte	in vivo –FJD rat and DMM mouse	Silencing by intra-articular injection attenuated cartilage destruction, reducing catabolic, hypertrophic and apoptotic marker expression.	[46]
mir-21-5p	antagomir	chondrocytes	in vivo—DMM mouse	Intra-articular injections to knockdown mir-21-5p significantly attenuated the severity of OA by modulating expression of FGF18.	[47]
miR-26a /26b	miRNA mimics	chondrocyte	in vivo—ACLT rat	Intra-articular over-expression attenuated development of OA in vivo and over-expression also promoted chondrocyte proliferation in vitro through the FUT4/NF-κB axis.	[48]
miR-320a	antisense oligonucleotides	chondrocytes	in vitro	Silencing reduced the IL-1β-mediated release of MMP13 and sGAG whilst enhancing expression of Col2a1 and ACAN.	[49]
miR-34a-5p	locked nucleic acid	chondrocytes and synovial fibroblasts	in vivo—DMM and high-fat diet/DMM mouse	Chondroprotective effects imparted following silencing in vivo by intra-articular injection. In vitro ASO treatment increased COL2A1 and ACAN in OA chondrocytes whilst reducing COL1A1 and TNFα in OA fibroblasts.	[50]
miR-449a	locked nucleic acid	cartilage and subchondral bone	in vivo—DMM rat	Intra-articular silencing promoted cartilage regeneration and expression of type II collagen and aggrecan in cartilage.	[51]
miR-98	pre-miRNA mimics	chondrocytes	in vitro	Overexpression significantly attenuated IL-1β-induced reduced TNFα production.	[45]
MK2	siRNA	chondrocytes	in vitro	MK2 silencing inhibited IL-1β-induced production of MMP3, MMP13 and PGE2.	[34,52]
MMP-13	siRNA	cartilage	in vivo—DMM mouse	Silencing alone or in combination with ADAMTS5 siRNA improved histological scores of OA severity in vivo by intra-articular injection.	[40]
p38	siRNA	chondrocytes	in vitro	P38 silencing inhibited IL-1β-induced production of MMP3, MMP13 and PGE2.	[34,52]
RDC1	siRNA	chondrocytes	in vitro	Silencing RDC1 in OA chondrocytes reduced expression of MMPs and hypertrophic markers.	[53]
YAP	siRNA	chondrocytes	in vivo—ACLT mouse	Silencing of YAP reduced chondrocyte apoptosis and inhibited IL-1β-induced catabolic factors. Intra-articular injection ameliorated OA development and reduced aberrant subchondral bone formation in vivo.	[54]
Subchondral Bone					
Target	Type of Oligonucleotide	Target Cell/Tissue	Study Model	Function	Ref.
DKK1	end-capped phosphorothioate antisense oligonucleotide	cartilage and subchondral bone	in vivo—ACLT and CIA rat	Oligonucleotides delivered intraperitoneally reduced disease severity, bone mineral density loss and reduced serum levels of bone resorption markers osteocalcin and CTX-1 and expression of TNFα, IL-1β, MMP3 and RANKL.	[55]
Leptin receptor long isoform (OBRl)	phosphorothioate double-stranded decoy oligonucleotide	osteoblasts	in vitro	Silencing abolished the leptin-mediated production of oncostatin M via miR-93/Akt signalling axis.	[56]
miR-29a	antisense oligonucleotides	subchondral mesenchymal stem cells	in vitro	Knockdown inhibits Wnt3a expression and impaired Wnt-mediated osteogenic differentiation via HDAC4.	[57]
Thyroid hormone receptor (THR)	siRNA	osteoblasts	in vivo—DMM mouse	THR knockdown downregulated THR regulatory genes including HIF-1α, VEGF and IGI-1. Intra-articular injection improved sclerotic subchondral bone formation, as well as an overall reduction in OA severity score.	[58]
Synovium					
Target	Type of Oligonucleotide	Target Cell/Tissue	Study Model	Function	Ref.
Dickkopf-1 (DKK1)	21-mer end-capped phosphorothioate antisense oligonucleotide	synovium	in vivo—ACLT rat	Intraperitoneally administered silencing reduced proteinases and angiogenic factors, reduced vessel distribution and formation and reduced cartilage injury.	[59]
FLIP	antisense oligo-deoxynucleotide	synovial fibroblasts	in vitro	FLIP knockdown increased fas-mediated apoptosis by 3-fold.	[60]
FoxC1	siRNA	synovial fibroblasts	in vivo—DMM mouse	Silencing inhibited IL-6, IL-8, TNF, ADAMTS-5, fibronectin, MMP3 and MMP13 and proliferation of OA synovial fibroblast, whilst intra-articular injection of FoxC1 siRNA prevented OA development in vivo.	[61]
Galectin-9	siRNA	synovial fibroblasts	in vitro	Galectin-9 knockdown increased apoptosis of human RA synovial fibroblasts.	[62]
Leptin receptor long isoform (OBRl)	antisense oligonucleotide	synovial fibroblasts	in vitro	Inflammatory OA fibroblast phenotype mediated by leptin was inhibited, thus reducing leptin-mediated IL-8 secretion via JAK2/STAT3 pathway.	[63]
Leptin receptor long isoform (OBRl)	phosphorothioate double-stranded decoy oligonucleotide	synovial fibroblasts	in vitro	Inflammatory OA fibroblast phenotype mediated by leptin was inhibited resulting in reduced IL-6 via IRS-1/PI3K/Akt/AP-1 pathway.	[64]
MALAT1	locked nucleic acid	synovial fibroblasts	in vitro	MALAT1 knockdown inhibited both the proliferative and inflammatory phenotype of obese OA synovial fibroblasts, resulting in reduced CXCL8 expression and secretion and increased expression of TRIM6, IL7R, HIST1H1C and MAML3.	[65]
Notch-1	antisense oligonucleotide	synovial fibroblasts	in vitro	Antisense Notch-1 oligonucleotide abrogated Notch-1 expression in the nucleus, preventing TNFα-induced translocation of Notch-1 intracellular domain (NICD) to the nucleus. Inhibits both basal and TNFα-induced proliferation of RA and OA synovial fibroblasts in a dose-dependent manner.	[66]
PTPN11 (SHP-2)	antisense knockdown	synovial fibroblasts	in vitro	Loss of SHP-2 inhibits migration, invasion, adhesion and survival of RA synovial fibroblasts through reduced PDGF-induced activation of MAPKs and upstream FAK.	[67]

ACLT, anterior cruciate ligament transection; CIA, collagenase-induced arthritis; DMM, destabilisation of the medial meniscus; FJD, facet joint degeneration; MIA, mono-iodoacetate-induced arthritis.

**Table 2 biomedicines-09-00902-t002:** Summary of studies optimising oligonucleotide delivery.

Target	Type of Oligonucleotide	Delivery Method	Study Model	Outcome	Ref.
Viral					
miR-128a/ATG12	antisense oligonucleotide	lentiviral	ACLT rat	Intra-articular administration of miR-128a antisense oligonucleotide disrupted ATG12 repression, stabilizing chondrocyte autophagy and delaying OA progression.	[43]
miR-101	antisense oligonucleotide	adenovirus	MIA rat	Antisense oligonucleotide silencing of miR-101 reduced cartilage degradation.	[42]
Biomaterial					
COX2	locked nucleic acid	hyaluronic acid hydrogel	human primary OA chondrocytes	Hydrogel-encapsulated LNA silenced COX2 over 14 days.	[68]
ADAMTS5	antisense oligonucleotide	fibrin and hyaluronic acid hydrogel	human primary OA chondrocytes	In a 3D two-gel cell construct, antisense oligonucleotides silenced ADAMTS5 at day 7 and 14 with concentrations of 5 and 10 µM.	[69]
Bioconjugation					
p38 MAPK	siRNA	cholesterol	L929 mouse fibroblast cell line and BALB/c mice	siRNA conjugated to cholesterol, TAT and pentratin silenced p38 MAPK in vitro whilst conjugation to cholesterol alone could circumvent immunostimulatory effects of TAT and pentratin in vivo.	[70]
myostatin	siRNA	cholesterol	CD-1 mice	A single intravenous injection of cholesterol-conjugated siRNA sustained myostatin silencing in skeletal muscle and in circulation over 21 days.	[71]
Nanoparticles					
GFP	morpholino antisense oligonucleotides	PEG-SWCNTs	DMM mice	Intra-articular injections of PEG-SWCNTs carrying anti-GFP oligonucleotides could silence GFP in GFP-transgenic mice	[72]
miR-141/200c	miRNA	PEG-PA	DMM mice	Intra-articular silencing of miR-141/200c has chondroprotective effects via downregulation of the IL-6/STAT3 pathway.	[73]

ACLT; anterior cruciate ligament transection, DMM, destabilisation of the medial meniscus; MIA, mono-iodoacetate-induced arthritis; PA, PEGylated polyamidoamine nanoparticle; PEG-SWCNTs, polyethylene glycol chain-modified single-walled carbon nanotube.

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
