# Peer review of "Oligonucleotide Therapies in the Treatment of Arthritis: A Narrative Review"

_biomedicines, 2021, doi:10.3390/biomedicines9080902_

Round 1

Reviewer 1 Report

The review is profound, important, and well organized.

Comment

  1. It would be interesting the authors opinion on the advantage of inhibition of the upregulated by disease genes, proteins, and signaling pathways using oligonucleotide therapies and downregulation of the same excessive activities by present treatment options.

Author Response

The review is profound, important, and well organized.

AUTHOR RESPONSE: We thank the reviewer for their kind comment.

It would be interesting the authors opinion on the advantage of inhibition of the upregulated by disease genes, proteins, and signalling pathways using oligonucleotide therapies and downregulation of the same excessive activities by present treatment options.

AUTHOR RESPONSE: We thank the reviewer for their suggestion and have addressed existing therapies and their limitations in the introduction and advantages of oligonucleotide therapies are addressed throughout and summarised in Figure.1B. We believe oligonucleotides have great promise as a potential therapeutic as mentioned in the concluding remarks.

“…the aim of current RA therapies is therefore to target the underlying disease pathology, a “treat-to-target” strategy to influence the disease course, not simply to reduce symptoms [9]. To this end, a number of disease-modifying anti-rheumatic drugs (DMARDs) have been approved for clinical use… In recent years improved understanding of the molecular pathology underlying RA disease has led to a revolution in RA treatment [18], with the emergence of a number of biological drugs designed to target specific disease-associated inflammatory cytokines (most notably TNF-α), and drugs targeting the activity of specific immune cell populations… Similar disease-modification has been achieved by biologics targeting other inflammatory immune processes… despite these successes, all these immunomodulatory medications are associated with adverse side effects and an increase in the risk of developing serious infections... Furthermore, with chronic treatment, efficacy can be greatly diminished... Therefore, there is still a great unmet medical need to develop more effective and safer RA therapeutics…”

“…Despite a number of clinical trials… there are no pharmacological drugs that have been proven to modify the disease course of OA (DMOADs) [35]… Therefore, patients are advised to take non-steroidal anti-inflammatory drugs… which have limited efficacy and are associated with toxicity over the long term.  Patients with knee OA can receive intra-articular corticosteroid injections into the diseased joint to alleviate pain and inflammation [27]. However, the benefit for patients is highly unpredictable… Furthermore, repeated steroid injections into the joint can exacerbate cartilage damage.  As a result, OA patients live with painful symptoms for several years before undergoing a joint replacement surgery, a procedure with a high percentage of patient dissatisfaction... To this end, the development of oligonucleotides, that modulate expression of targets at an upstream level, where functional protein products are not made, may offer the potential to target pathological processes across multiple cell types including chondrocytes in the cartilage, osteoblasts in the bone and fibroblasts in the synovium…”

Reviewer 2 Report

Review: “Oligonucleotide therapies in the treatment of inflammatory  joint disease”

The focus of the review is interesting.

My comments are as follow:

Lines 24-28: the authors reported that OA is a disease of the whole joint, including not only the loss of articular cartilage mass, but also subchondral bone sclerosis and inflammation to the synovium. However, the authors should also mention the role of meniscus and ligaments in OA and the recent research on the involvement of the infrapatellar fat pad in OA, which is fibrotic and inflamed (doi.org/10.1002/art.34453; doi: 10.3390/ijms21176016).

A brief paragraph on the methods used for this review (Specify all databases, and registers used; keywords used; specify the methods used to decide whether a study met the inclusion criteria of the review; and specify the selected time range of the search) should be added.

Figure 1 should be improved. In particular, all the joint tissues are involved in OA, not only cartilage, bone and synovium.

In the section 2 of the review, the authors describe synovial inflammation in OA and RA and discuss about fibroblast-like synoviocytes (FLS), also termed synovial fibroblasts. However, synovial membrane is composed by two types of synoviocytes, macrophagic cells (type A cells) and fibroblast-like cells (type B cells).

It is not clear why the authors reported studies focused on oligonucleotides impacting synoviocytes apoptosis in section 2, where the authors the authors are focused on oligonucleotides targeting synovial inflammation.

Line 20, pag.4: “bene” should be corrected.

Lines 17-25, pag.4: The text should be better organized. The authors reported twice information about lncRNA MALAT1.

Lines 28-29, pag.4: the authors reported “As yet, few studies have reported the effect of oligonucleotides designed to target the synovium in preclinical in vivo models of either RA or OA disease.” and then, they cited only two studies on OA. Are there any other studies on OA? Is there any study on RA?

Lines 42-43, pag. 4: “Conversely, in OA the subchondral becomes sclerotic,…” . The authors should specify “subchondral bone”.

Lines 31-32, pag.5: pro-inflammatory cytokines are secreted not only from synovial fibroblasts but also from other OA tissues such as infrapatellar fat pad.

Lines 35-36, pag.5: “A key component in this is the cellular cross-talk between synovial fibroblasts and chondrocytes,” This part of the sentence should be rewritten.

The authors discussed about oligonucleotides targeting the different tissues (cartilage, synovial membrane and subchondral bone) reporting also in vivo studies. Then, they reported a section entitled “Optimisation of Oligonucleotide Delivery into The Joint Tissues” focused on in vivo studies. This is confusing for the reader. It would be useful to move all the in vivo studies in the section 5, while reporting in vitro studies in sections 2,3 and 4. A table should be added for section 5.

Table 1 should be better organized reporting studies related to oligonucleotides for each tissue analyzed. I suggest to divide the table according to the different tissue/cell targeted to avoid confusion reflecting the sections of the review.

It is unclear whether clinical trials are ongoing. This point should be better discussed.

There are several parts of the review without references: Lines 37-42, pag.4; Line 53 pag.4 and line 1 pag.5; Lines 23-30, pag.5; Lines 34-50.

Abbreviations should be defined at first mention and use consistently throughout the manuscript (for example OA and RA). The authors should check.

Author Response

The focus of the review is interesting.

AUTHOR RESPONSE: We thank the reviewer for their insightful comments and have made corrections as detailed below.

Lines 24-28: the authors reported that OA is a disease of the whole joint, including not only the loss of articular cartilage mass, but also subchondral bone sclerosis and inflammation to the synovium. However, the authors should also mention the role of meniscus and ligaments in OA and the recent research on the involvement of the infrapatellar fat pad in OA, which is fibrotic and inflamed (doi.org/10.1002/art.34453; doi: 10.3390/ijms21176016)

AUTHOR RESPONSE: We agree with the reviewer and have updated the sentence and references to reflect this and to now read “OA is a disease of the whole joint, including not only the loss of articular cartilage mass and meniscal and/or ligament damage, but also subchondral bone sclerosis and inflammation of the synovium and infrapatellar fat pad [29-32].”

REVIEWER #1 COMMENT: review (Specify all databases, and registers used; keywords used; specify the methods used to decide whether a study met the inclusion criteria of the review; and specify the selected time range of the search) should be added.

AUTHOR RESPONSE: We have included a methods section (Section 2) detailing the relevant information.

Figure 1 should be improved. In particular, all the joint tissues are involved in OA, not only cartilage, bone and synovium.

AUTHOR RESPONSE: Figure and legend has been amended to reflect all joint tissues involved.

In the section 2 of the review, the authors describe synovial inflammation in OA and RA and discuss about fibroblast-like synoviocytes (FLS), also termed synovial fibroblasts. However, synovial membrane is composed by two types of synoviocytes, macrophagic cells (type A cells) and fibroblast-like cells (type B cells).

AUTHOR RESPONSE: We agree with Reviewer 1’s comment, also the term “FLS” is used interchangeable throughout much of OA and RA literature to describe synovial fibroblasts. Here, we have stipulated that FLS are also termed synovial fibroblasts to highlight the focus on this cell type.

It is not clear why the authors reported studies focused on oligonucleotides impacting synoviocytes apoptosis in section 2, where the authors the authors are focused on oligonucleotides targeting synovial inflammation.

AUTHOR RESPONSE: This sentence has been amended to clarify the link to inflammation:

“More recently, susceptibility of human OA and RA synovial fibroblasts to fas-mediated apoptosis was increased by antisense oligonucleotides targeting the anti-apoptotic gene FLICE-inhibitory protein (FLIP) [49], and increased apoptosis of human RA synovial fibroblasts was induced upon antisense oligonucleotide targeting of galectin-9 [50]. These targets, unsilenced, provide protection against apoptosis thus maintaining fibroblast populations contributing to persistent inflammation and as such may be valuable targets in combating synovial inflammation and hyperplasia.”

Line 20, pag.4: “bene” should be corrected.

AUTHOR RESPONSE: This has been corrected to “been”.

Lines 17-25, pag.4: The text should be better organized. The authors reported twice information about lncRNA MALAT1.

AUTHOR RESPONSE: This has been amended to read:

“LncRNAs are a relatively novel class of non-coding RNAs, which have been shown to regulate gene expression at both the epigenetic pre-transcriptional and post-transcriptional level through their ability to act as scaffolds for the binding of proteins and other RNAs [55-59]. Recently, the MALAT1 lncRNA was found to regulate the inflammatory response of articular OA chondrocytes [60] and synovial fibroblasts [46].  A locked nucleic acid (LNA) oligonucleotide targeting the MALAT1 lncRNA was found to inhibit both the proliferative and inflammatory phenotype of obese OA synovial fibroblasts [46].”

Lines 28-29, pag.4: the authors reported “As yet, few studies have reported the effect of oligonucleotides designed to target the synovium in preclinical in vivo models of either RA or OA disease.” and then, they cited only two studies on OA. Are there any other studies on OA? Is there any study on RA?

AUTHOR RESPONSE: Thank you for bringing this to our attention we have now updated the sentence with missing references for both OA and RA.“…As yet, few studies have reported the effect of oligonucleotides designed to target the synovium in preclinical in vivo models of either RA or OA disease [63-68]…”

Lines 42-43, pag. 4: “Conversely, in OA the subchondral becomes sclerotic,…” . The authors should specify “subchondral bone”

AUTHOR RESPONSE: This has been corrected: “Conversely, in OA the subchondral bone becomes sclerotic…”

Lines 31-32, pag.5: pro-inflammatory cytokines are secreted not only from synovial fibroblasts but also from other OA tissues such as infrapatellar fat pad.

AUTHOR RESPONSE: Amended to read: “with pro-inflammatory cytokines secreted by various joint tissues…”

Lines 35-36, pag.5: “A key component in this is the cellular cross-talk between synovial fibroblasts and chondrocytes,” This part of the sentence should be rewritten.

AUTHOR RESPONSE: This sentence has been rewritten to read: “The cellular cross-talk between synovial fibroblasts and chondrocytes is fundamental in this …”

The authors discussed about oligonucleotides targeting the different tissues (cartilage, synovial membrane and subchondral bone) reporting also in vivo studies. Then, they reported a section entitled “Optimisation of Oligonucleotide Delivery into The Joint Tissues” focused on in vivo studies. This is confusing for the reader. It would be useful to move all the in vivo studies in the section 5, while reporting in vitro studies in sections 2,3 and 4. A table should be added for section 5.

AUTHOR RESPONSE: To clarify, the section focus is on studies that have optimised oligonucleotide method of delivery (using viral vectors, various biomaterials, bioconjugation to penetrating peptides, nanoparticles etc) and as such the in vivo studies mentioned in other sections do not fit this focus. We have amended the title of this section to avoid confusion: “Optimisation of oligonucleotide method of delivery into joint tissues”. Additionally, Table.2 has been added to summarise studies mentioned in this section (now section 6).

Table 1 should be better organized reporting studies related to oligonucleotides for each tissue analyzed. I suggest to divide the table according to the different tissue/cell targeted to avoid confusion reflecting the sections of the review.

AUTHOR RESPONSE: Table 1 has been divide to reflect the sections in the review.

It is unclear whether clinical trials are ongoing. This point should be better discussed.

AUTHOR RESPONSE: This has been clarified in the conclusion. “Although yet to be successfully tested in clinical trials for arthritis treatment, data from preclinical experimental…”

There are several parts of the review without references:

AUTHOR RESPONSE: We thank the reviewer for bringing this to our attention and have added the relevant references as follows.

Lines 37-42, pag.4;

AUTHOR RESPONSE: “In bone homeostasis, receptor activator of nuclear factor kappa B (RANK)/RANKL pathway activates NF-kB induced transcription factors that provides the balance between bone resorption and bone formation [65,66]. However, this homeostasis is lost in the pro-inflammatory microenvironment within the RA joint, where there is the promotion of bone erosion via the pro-inflammatory cytokine induction of RANKL, which binds to the RANK receptor on osteoclasts and activates their bone resorbing activity [65].”

Line 53 pag.4 and line 1 pag.5;

AUTHOR RESPONSE: “In vitro, antisense oligonucleotides have also been shown to effectively impact on osteogenic differentiation [76].”

Lines 23-30, pag.5;

AUTHOR RESPONSE: “This is driven largely by a pathological switch in the phenotype of chondrocytes [35]. In healthy adult cartilage, the chondrocytes are embedded in the extracellular matrix and are in a relatively metabolically inactive state, where they produce type II collagen and aggrecan proteoglycans that hydrate the cartilage and provide the cartilage with its load-absorbing properties [83]. However, in OA the chondrocytes proliferate and become hypertrophic, switching from producing extracellular matrix proteins to producing catabolic MMPs and aggrecanases ADAMTS4 and ADAMTS5, which degrade type II collagen and aggrecan proteoglycan respectively [35,83].”

Lines 34-50, pag 6.

AUTHOR RESPONSE: “…. the articular cartilage represents a formidable challenge since it is avascular, and thus unmodified drug delivery into the chondrocytes occurs slowly and inefficiently [99]. Oligonucleotide therapeutics, at approximately 20 bases, are larger than most small molecule drugs and are also anionic, this making diffusion across a negatively-charged cell membrane challenging [99,100] .... Furthermore, delivery of the therapeutic locally to the joint limits the risk of unwanted side effects at distal sites [99,100].”

Abbreviations should be defined at first mention and use consistently throughout the manuscript (for example OA and RA). The authors should check.

AUTHOR RESPONSE: Thank you for bringing this to our attention, the abbreviations have been revised and amended where required.

Reviewer 3 Report

This topic is very interesting and important in clinical applications.

Title: identify RA and OA: " A narrative review ", rather than present inflammatory joint disease, which is too vague.

The more detailed information and current related references should be added, For example,

-Please add the information of oral/intra-articular injection of glucosamine-based biomaterials (e.g, glucosamine sulfate, glucosamine hydrochloride, N-acetylglucosamine) and platelet-rich plasma  in the OA part. 

-Please provide the cellular pathways mechanisms. Presenting a graph may be suggested. 
-Please discuss "delivery challenges" and "safety concerns" for clinical application.
-Summarize a table strengths and limitations. Currently, it is too many words to make readers recognize clearly

Other comments:

Figure 1 is good. 

Table 1. Mode in vivo. Please clarify which kind of animal usage.

Both RA and OA are chronic  inflammatory conditions. However, RA is a systematic problem; OA is relatively local problem. Does the oligonucleotide therapy strategy show different? If yes, Please point out. 

-Suggest to read the following references,

Nature Reviews Drug Discovery volume 19, pages 673–694 (2020)

Bone. 2020 Sep;138:115461

Author Response

This topic is very interesting and important in clinical applications.

AUTHOR RESPONSE: We thank the reviewer for their comments which have been addressed below.

Title: identify RA and OA: " A narrative review ", rather than present inflammatory joint disease, which is too vague.

AUTHOR RESPONSE: The title has been amended as follows:“Oligonucleotide therapies in the treatment of arthritis: a narrative review”

The more detailed information and current related references should be added, For example,

-Please add the information of oral/intra-articular injection of glucosamine-based biomaterials (e.g, glucosamine sulfate, glucosamine hydrochloride, N-acetylglucosamine) and platelet-rich plasma in the OA part. 

AUTHOR RESPONSE: References have been added to section 1.2 to reflect current treatment options: “Additionally, platelet-rich plasma and glucosamine-based supplementation, although show some promise for alleviating pain and delaying OA progression, these effects have been highly variable and debated in trials [36,37].”

-Please provide the cellular pathways mechanisms. Presenting a graph may be suggested. 

AUTHOR RESPONSE: We thank the reviewer for this suggestion and have detailed an overview of the cellular mechanisms in Figure.1A, which is discussed in the figure legend and also addressed throughout the review as exampled below:

“…cycle of inflammation and cartilage damage via activation of TLRs, the MAPK pathway and the NF-kB pathway…”

“... provided chondroprotection when delivered by intra-articular injection into a DMM murine experimental OA model via inhibition of IL-6/STAT3 pathway…”

“… it is known to inhibit NF-kB activity and thus the production of inflammatory cytokines including tumour necrosis factor alpha (TNF-α) …”

“… receptor activator of nuclear factor kappa B (RANK)/RANKL pathway activates NF-kB induced transcription factors that provides the balance between bone resorption and bone formation…”

-Please discuss "delivery challenges" and "safety concerns" for clinical application.

AUTHOR RESPONSE: We have addressed the challenges and safety in section 6 and have also summarized this in Figure.1B.

“…One of the central challenges in the development of drugs to treat inflammatory joint diseases such as OA and RA is the effective delivery of the therapeutic to the diseased tissues of the joint. In particular, the articular cartilage represents a formidable challenge since it is avascular, and thus unmodified drug delivery into the chondrocytes occurs slowly and inefficiently [103]. Oligonucleotide therapeutics, at approximately 20 bases, are larger than most small molecule drugs and are also anionic, this making diffusion across a negatively-charged cell membrane challenging [103,104]…”

“…the ability to deliver therapeutics, including oligonucleotide therapeutics, directly into the joint space via intra-articular injection, thus maximising the ability of the oligonucleotides to target the diseased tissues, is an advantage…”

“…delivery of the therapeutic locally to the joint limits the risk of unwanted side effects at distal sites…”

“…clinically there is still concern about whether viral-mediated delivery of therapeutics will lead to toxicity…”

“…. Although confirmation in the clinical setting is still required, such nanoparticulated drugs (nanodrugs) have recently gained recognition…nanoparticles can exploit the synovial capillary networks within the synovial tissue and previous studies have demonstrated that even systemic delivery of nanodrugs can be detected in the synovial joint in experimental models…”

“…since oligonucleotide therapeutics are based on gene sequences, they are expected to act specifically on the target gene, and thus may be considered less likely to have off-target effects and to elicit adverse side effects...

-Summarize a table strengths and limitations. Currently, it is too many words to make readers recognize clearly

AUTHOR RESPONSE: We thank the reviewer for an excellent suggestion. The strengths and limitations of oligonucleotide therapies have been summarised in Figure.1B.

REVIEWER #2 COMMENT: Figure 1 is good.

AUTHOR RESPONSE: We thank the reviewer for their kind comment. 

Table 1. Mode in vivo. Please clarify which kind of animal usage.

AUTHOR RESPONSE: Animals used in studies have been clarified in Table 1.

Both RA and OA are chronic inflammatory conditions. However, RA is a systematic problem; OA is relatively local problem. Does the oligonucleotide therapy strategy show different? If yes, Please point out

AUTHOR RESPONSE: No, studies have investigated oligonucleotide distribution post intravenous delivery in rodent joints, and find these oligonucleotides do not reach avascular tissues like the cartilage or the poorly fenestrated capillaries of OA synovium. Hence, joint localised delivery strategy is thought to be beneficial over systemic targeting allowing increased efficiency of drug delivery to target joint tissues. We have discussed methods of delivery and modifications which may allow for better systemic delivery in section 6.

Suggest to read the following references,

Nature Reviews Drug Discovery volume 19, pages 673–694 (2020)

Bone. 2020 Sep;138:115461

AUTHOR RESPONSE: Addressed as part of missing references identified by Reviewer 1.

Round 2

Reviewer 2 Report

No additional comments. 

I only noticed that in figure 1, which is divided in a and b, letter a is missing. 

Reviewer 3 Report

Congratulations. Thank you for the revision.